# The Active and Noise-Tolerant Strategic Perceptron

## Abstract

Strategic classification is an emerging area of modern machine learning research that models scenarios where input features are provided by individuals who might manipulate them to receive better outcomes, e.g., in hiring, admissions, and loan decisions. Prior work has focused on supervised settings, where human experts label all training examples.

However, labeling all training data can be costly, as it requires expert intervention. In this work, we initiate the study of active learning for strategic classification, where the learning algorithm takes a much more active role compared to the classic fully supervised setting in order to learn with much fewer label requests.

Our main result provides an algorithm for actively learning linear separators in the strategic setting while preserving the exponential improvement in label complexity over passive learning previously achieved in the simpler non-strategic case. Specifically, we show that for data uniformly distributed over the unit sphere, a modified version of the Active Perceptron algorithm [Dasgupta et al., 2005, Yan and Zhang, 2017], can achieve excess error $\varepsilon$ after requesting only $\tilde{O}\left(d \ln \frac{1}{\varepsilon}\right)$ labels and making an additive $\tilde{O}\left(d \ln \frac{1}{\varepsilon}\right)$ mistakes compared to the best classifier, when the $\tilde{\Omega}(\varepsilon)$ fraction of the inputs are flipped. These algorithms are computationally efficient with number of label queries substantially better than prior work in strategic Perceptron [Ahmadi et al., 2021] under distributional assumptions.

## 1 Introduction

**Overview** We initiate the study of active learning algorithms that classify strategic agents. Active learning is a well-established framework within machine learning that selectively queries labels, allowing algorithms to achieve higher accuracy and efficiency than traditional supervised learning methods. This is particularly useful when labeling data is expensive or time-consuming, which includes canonical examples in strategic classification such as hiring, admissions, and loan lending. Strategic classification addresses the challenge of learning classification rules when the data provided by agents is not truthful. In these scenarios, agents modify their feature vectors to appear more favorable to the classifier, typically in pursuit of a positive classification outcome. This manipulation introduces additional obstacles beyond the standard problems in learning accurate classification rules from true data. The goal of this research is to develop active and noise tolerent algorithms in strategic settings, that is classification algorithms that can efficiently and accurately classify strategic agents while minimizing the number of queries for labels. The challenge lies in simultaneously addressing the strategic manipulation of data and optimizing the learning process to require fewer labeling requests, thus improving efficiency.

Active learning algorithms work on the premise that we can learn by only obtaining the labels of a few select very informative examples. Typically, the decision of whether to request the label of an example or not is based on the features of that example. Since we consider the strategic setting

where the features might have been manipulated, there is a danger that we end up asking for labels of examples that might not be so informative after all, therefore derailing the active learning process and obtaining classifiers that do badly under original data distribution. In this work, nonetheless, we overcome these challenges in several important cases by showing how to make existing active learning algorithms robust, using properties of the learned classifiers to update only on points that can be guaranteed to be manipulated.

Furthermore, ideas from active learning have been transformative in improving the design of passive learning algorithms. In several challenging learning problems, where existing passive learning methods failed to achieve optimal guarantees, incorporating techniques from active learning has led to significant breakthroughs [Awasthi et al., 2014, 2015, 2016, 2017]. In this work, we demonstrate how active learning principles can overcome fundamental limitations in classifying strategic entities. Specifically, we show that while prior passive learning algorithms struggled in this setting, selectively ignoring certain labels and focusing on informative queries leads to stronger theoretical guarantees and more robust performance.

## 1.1 Setup

We consider an online linear classification problem in which the individuals being classified are strategic, as in Ahmadi et al. [2021]. Each individual arriving at the classifier wishes to be classified positively and, if necessary, will manipulate their feature vector to achieve this outcome. More formally, an individual's true feature vector is $z_t$, but they may choose to report a manipulated vector $x_t$ if it results in receiving a positive classification. The manipulation comes at a cost, which reflects how far their reported vector $x_t$ is from their true vector $z_t$.

Specifically, we model individuals as utility-maximizing agents, where the utility is defined as the value received from the classification outcome minus the cost of manipulation. If an individual is classified as positive, their value is 1, otherwise, it is 0. Thus, the goal of each individual is to maximize:

$$\max_{x_t}[\text{value}(x_t) - \text{cost}(z_t, x_t)],$$

where $\text{value}(x_t) = 1$ if the manipulated vector $x_t$ is classified as positive and 0 if classified as negative. The cost function $\text{cost}(z_t, x_t)$ quantifies the cost of manipulating the features from $z_t$ to $x_t$. In this setting, if an individual can manipulate their features at a cost of at most 1 to change their classification from negative to positive, they will do so in the least costly way; otherwise, they will not manipulate their features.

We consider a more challenging setting than Ahmadi et al. [2021], namely the active learning setting, where we only aim to ask for labels of selected samples in order to minimize the need for human intervention. In such active learning scenarios, even in the simpler non-strategic setting, distributional assumptions are needed to provably show improvements in label complexity in active scenarios over non-active ones [Dasgupta et al., 2005, Balcan et al., 2007, 2006, Balcan and Urner, 2014, Hanneke, 2014]. The most widely studied distributional assumption is that the feature vectors $z_t$ are uniformly distributed, which is the setting we consider in this paper. Formally, we assume that the true feature vectors $z_t$ of individuals are uniformly distributed within a $d$-dimensional unit ball centered at the origin. In the realizable case, there exists a true classifier $u$ such that $u \cdot z_t \geq 0$ for all positively labeled points and $u \cdot z_t < 0$ for all negatively labeled points. In the nonrealizable case, we extend this setting to allow for some fraction of points that do not strictly conform to this separation. Specifically, a certain proportion of points may have labels inconsistent with the best homogeneous linear classifier $u$, reflecting noise in the data.

The task for the learning algorithm is to overcome these manipulations and accurately classify the true features, $z_t$, while minimizing the number of labeling queries. By integrating active learning, we aim to limit the number of labels requested from a costly oracle, thereby reducing the labeling effort needed to achieve high classification accuracy. Additionally, we aim to develop algorithms that are robust against strategic manipulation, ensuring that agents cannot easily game the system to receive positive classifications unjustly.

One of the main challenges in this setup is designing classifiers that can operate effectively with manipulated data, without relying on access to the true features of agents. Traditional active learning algorithms do not account for such strategic behavior, which makes them vulnerable to manipulation.

Moreover, the online nature of the problem introduces additional difficulties, as the classifier must adapt in real-time to changing behaviors from the agents and evolving data distributions.

By studying the interplay between active learning and strategic behavior, we aim to provide a new class of learning algorithms that efficiently learn from strategically manipulated data with minimal labeling requests. These algorithms will have broad applications, from financial systems where individuals manipulate credit scores, to online platforms where users alter their behavior to achieve better outcomes.

## 1.2 Technical Contributions

This work addresses several critical challenges in active learning for strategic classification, advancing the state of the art in handling strategic behavior, noise, and realistic data distributions. When the examples are drawn from a uniform distribution over a unit sphere, our contributions are summarized as follows:

1. **Active Strategic Classification:** We extend active learning guarantees from non-strategic to strategic settings, providing theoretical and algorithmic foundations for robust classification in the presence of manipulation. In the realizable case, our results imply we can achieve genralization error $\varepsilon$ after requesting $\tilde{O}\left(d\ln\frac{1}{\varepsilon}\right)$ labels and making $\tilde{O}\left(d\ln\frac{1}{\varepsilon}\right)$ mistakes.

2. **Noise in Strategic Classification:** In the nonrealizable case, can achieve excess error $\Theta(\varepsilon)$ after requesting only $\tilde{O}\left(d\ln\frac{1}{\varepsilon}\right)$ labels and making an additive $\tilde{O}\left(d\ln\frac{1}{\varepsilon}\right)$ mistakes compared to the best classifier, when the $\tilde{\Omega}(\varepsilon)$ fraction of the inputs are flipped. We resolve an open problem posed by Ahmadi et al. [2021] regarding handling noise in classification, moving beyond perfect separability. Previous techniques were insufficient for addressing this issue.

## 1.3 Adapting Algorithms for Strategic Settings

Our main technical contribution is to integrate techniques from strategic classification and active learning. Remarkably, we show how ideas that were developed for noise tolerance of active learning algorithms and were not originally designed for strategic settings can be adapted in the strategic setting. These adaptations leverage the behavior of utility-maximizing strategic agents. Notably, once these adaptations are made, the main steps of the proof remain similar.

**Threshold adjustment for positive classification.** A key adaptation in the algorithm is the introduction of a positive threshold for the dot product with the classifier's weight vector to determine positive classification. In other words, we raise the bar for a point to be classified as positive. To illustrate, consider the case where the original data is separable. In the early phases, when the classifier's direction may significantly deviate from the true one, this threshold may not be optimal. However, as the algorithm converges, this adjustment ensures that truly positive points are either already on the correct side of the boosted threshold or can manipulate to reach it. For negative points, this modification imposes a cost too high to justify manipulation. A similar analysis holds even when the data is not fully separable but contains a limited amount of noise. Although a similar modification was applied successfully in Ahmadi et al. [2021] in the realizable setting, this adjusted classification rule could not be integrated with an appropriate update rule in the nonrealizable setting.

**Focusing label-queries on unmanipulated examples.** Prior results on active learning define label-requesting regions and only query examples in that region. The algorithm we propose requests labels only for points classified as negative and within the label-requesting region. The key observation is that, by the nature of utility-maximizing agents, any point classified as negative remains unmanipulated. Thus, these examples reflect their true positions and provide reliable information. Additionally, assuming a uniform distribution, these queried examples serve as a representative set of all examples in the label-requesting region. We note that similar modification was done in Yan and Zhang [2017], however it was introduced in a different context and not motivated by strategic considerations.

A crucial insight guiding this approach is the symmetry in the classification process. Consider the set of misclassified points. Similar to the non-strategic setting, our algorithm is designed such that, in expectation, the number of truly positive points misclassified as negative is equal to the number of

negatives misclassified as positive. The key fact is that by ignoring the half that has been misclassified as positive and only considering those misclassified as negative, the algorithm still achieves similar guarantees.

Why is this beneficial? The key reason is that points misclassified as negative have not been manipulated, making them suitable for use in the update steps. But why is it acceptable to ignore points misclassified as positive? Intuitively, for hyperplanes crossing the origin, every positive point on one side has a corresponding negative point with opposite coordinates. Observing one provides the same directional information about the true classification vector as the other. Since both contribute identically to the update process in Perceptron-based algorithms as well as Dasgupta et al. [2005] and Yan and Zhang [2017]'s modification, we can confidently focus solely on points misclassified as negative.

**Beyond uniform distribution.** Although we focus on the uniform distribution for simplicity, the ideas extend more generally. Our results do not rely strictly on uniformity and still hold, with diminished guarantees, as long as the underlying distribution is smooth enough—specifically, when the probability density of points along any given ray is scaled by a fixed constant relative to any other ray. While we do not formally prove this, the key insights remain valid under such smoothness conditions.

## 1.4 Related Literature

Active learning has a long-standing history in machine learning [Balcan and Urner, 2014]. The central idea is that a learner can achieve better generalization with fewer labeled examples by selectively querying the most informative ones. Settles [2012] provides an extensive survey of active learning algorithms and their applications, highlighting the potential efficiency gains of this approach. From a theoretical perspective, key paradigms and analysis frameworks include disagreement-based active learning, first studied in the presence of noise by Balcan et al. [2006], and further developed by many others [Dasgupta et al., 2007, Koltchinskii, 2010, Beygelzimer et al., 2010, Hanneke, 2007]. Another widely studied and more practical paradigm is margin-based active learning, where the algorithm queries only points near the current decision boundary. Our work falls into this category [Yan and Zhang, 2017, Dasgupta et al., 2005, Balcan et al., 2007, Balcan and Long, 2013, Awasthi et al., 2014], and is most closely related to Yan and Zhang [2017], as discussed in Section 3.1.

The study of strategic classification has gained increasing attention in recent years, motivated by the need to understand how individuals or entities may "game" machine learning systems. Hardt et al. [2016] introduced foundational models for strategic classification, where individuals manipulate their feature vectors to obtain more favorable outcomes. This line of work has since been extended to examine how classifiers can be designed to be robust to such manipulations, including contributions by Brückner et al. [2012] and Milli et al. [2019]. Several other works explore variations of the strategic classification model, including Dong et al. [2018], Braverman and Garg [2020], Harris et al. [2023], but all rely on fully labeled training data. Strategic classification is typically motivated by applications such as admissions, hiring, and financial decision-making (e.g., loan lending), where individuals have incentives to modify their input data. In all of these domains, acquiring labeled data—required in all prior work—is often costly, as it typically involves human expert judgment. This highlights the importance of developing algorithms that can learn effectively with fewer labeled examples.

Prior work on online binary ($\pm 1$) classification in strategic settings—aimed at optimizing classification accuracy—has primarily focused on two cases: (i) when the original data is perfectly separable, as in Ahmadi et al. [2021], who showed that guarantees achievable in this setting can fail under even slight inseparability (noise); and (ii) when the data is not separable, but the algorithm's performance degrades arbitrarily with the level of noise, as in Chen et al. [2020]. In both cases, prior work falls short of providing robust guarantees in the presence of moderate noise, especially while maintaining low label complexity.

The intersection of active learning and strategic classification is a relatively new area of research. Most active learning models assume truthful data, whereas strategic classification assumes that agents may manipulate their features. The challenge we address is how to integrate active learning techniques in the context of strategic agents who provide manipulated data, thus balancing the need for efficient learning with robustness against manipulation.

## 2 Model and Preliminaries

**Strategic Manipulation and Utility Model.** We study an online classification problem where a sequence of examples in $\mathbb{R}^d$ arrives one at a time. Each example corresponds to an individual with $d$ attributes, who wishes to be classified positively. Individuals have the ability to manipulate their attributes at some cost. Let $z_t$ denote the true, unmanipulated instance vector of the $t$-th individual, and let $x_t$ be the reported (potentially manipulated) vector observed by the classifier.

We consider two settings:

- Realizable Case: There exists a true classifier $u$ such that all positive examples satisfy $u \cdot z_t \geq 0$, and all negative examples satisfy $u \cdot z_t < 0$.

- Non-Realizable Case: Some fraction of examples may have labels inconsistent with the classifier $u$, introducing label noise.

We assume individuals are utility-maximizing agents who manipulate their attributes to achieve a positive classification while minimizing manipulation cost. Each individual derives a value of 1 if classified as positive and 0 otherwise. The cost of manipulation, denoted as $\text{cost}(z_t, x_t)$, quantifies the effort required to modify $z_t$ to $x_t$. The individual's goal is to maximize:

$$\max_{x_t}[\text{value}(x_t) - \text{cost}(z_t, x_t)].$$

If manipulation is possible within cost constraints, the agent *moves*, i.e., changes its feature vector to the cheapest point that ensures a positive classification. Otherwise, they *remain* at $z_t$, i.e., do not manipulate.

We consider the following setting for the Euclidean cost function, where the cost is proportional to the $\ell_2$ distance between $z_t$ and $x_t$, i.e., $\text{cost}(z_t, x_t) = c\|x_t - z_t\|_2$. Here, $c$ represents the per-unit movement cost.

**Instance Space and Distributional Assumptions.** We denote the instance space by $\mathcal{Z}$ and the label space by $\mathcal{Y}$. The true feature vectors $z_t$ belong to instance space $\mathcal{Z} = \{z \in R^d : \|z\| \leq 1\}$; a unit $d$-dimensional ball. The label space $\mathcal{Y} = \{+1, -1\}$. We assume all examples $z$ are drawn i.i.d. from the uniform distribution $D$ over $\mathcal{Z}$. Upon sampling an example, our algorithm observes $x$, whose true instance vector $z \in \mathcal{Z}$ is drawn from $D$ and whose label is hidden by default. Our algorithm is allowed to make queries to a labeling oracle $\mathcal{O}$, which returns the true label for $z$. In line with prior work [Dasgupta et al., 2005, Yan and Zhang, 2017] on nonstrategic settings, the goal of the learning algorithm is to classify the true instance vectors accurately while minimizing the number of label queries. To achieve this, we leverage active learning techniques, which allow querying labels only when necessary, reducing reliance on labeled data from a costly oracle.

In the nonrealizable setting, where there may not be a homogeneous halfspace including all $+1$ and excluding all $-1$ examples, we consider a bounded inseparability (noise) measure $\nu$. Specifically, we say that our setting satisfies the $\nu$-bounded inseparability (noise) condition for some $\nu \in [0, 1]$ with respect to $u$, if $\mathbb{P}[Y \neq \text{sign}(u \cdot Z)] \leq \nu$.

The output of our algorithm is a unit-norm vector $w$ defining a halfspace of the form $w \cdot x \geq b$, where $b \geq 0$. That is, the resulting halfspace is not necessarily homogeneous. We define the error rate of the halfspace $h$ as $\text{err}(h) = \mathbb{P}[\mathbb{1}(w \cdot X \geq b) \neq Y]$.

For any two vectors $w_1, w_2$, let $\theta(w_1, w_2) = \arccos(v_1 \cdot v_2)$ be the angle between them.

Our algorithm uses norm-1 scaled version of the observed examples for the update function. We use the following definition to denote the norm-1 scaled examples.

**Definition 1 ($\hat{x}$).** *For any non-zero $d$-dimensional vector $x$, we define $\hat{x}$ as its scaled version whose length is equal to 1; i.e., $\hat{x} = \frac{x}{\|x\|}$.*

**Online Learning Setting and Learning Objective.** Our goal is to design an efficient algorithm such that with probability at least $1 - \delta$, outputs a halfspace whose error is at most $\varepsilon$ larger than $u$ for $\mathcal{Z}$. We require the algorithm to be efficient, use a the minimal number of label queries, and make at most $\Theta(\varepsilon)$ mistakes.

We assume that the examples arrive online. Each example is an input provided by a strategic agent. The agent knows the current prediction rule. The algorithm has access to the manipulation cost. Given the current prediction rule, the agent selects a utility maximizing action $\boldsymbol{x}$. The algorithm observes the potentially manipulated example $\boldsymbol{x}$. Upon a label query, the algorithm receives the true label of the example.

# 3 Active and Noise-Tolerant Strategic Perceptron

In this section, we overcome the challenges of designing active learning algorithms in strategic settings. We propose a modified active Perceptron algorithm that adapts to strategic behavior by selectively querying labels and leveraging the unmanipulated nature of certain points. The modifications ensure that the algorithm remains robust to strategic actions while maintaining the efficiency of active learning.

The proposed algorithm includes key changes to handle strategic manipulations by agents. It focuses on querying only examples classified as negative, using the fact that these points are not manipulated because agents gain no extra by modifying them. This approach ensures that updates are made using true, unaltered data. Other adjustments, such as scaling examples to fit on the unit ball and increasing the classification threshold, help the algorithm stay accurate and require a minimal number of label queries despite the strategic behavior of agents.

**Theorem 2.** *Suppose Algorithm 1 has inputs satisfying the $\nu$-bounded inseparability condition with respect to halfspace $\boldsymbol{u}$, initial halfspace $\boldsymbol{v}_0$ such that $\theta(\boldsymbol{v}_0, \boldsymbol{u}) \leq \pi/2$, target error $\varepsilon$, confidence $\delta$, sample schedule $\{m_k\}$ where $m_k = \Theta\left(d\left(\ln d + \ln \frac{k}{\delta}\right)\right)$, and band width $\{b_k\}$ where $b_k = \Theta\left(\frac{2^{-k}}{\sqrt{d}\ln(km_k/\delta)}\right)$. Additionally, $\nu \leq \Theta\left(\frac{\varepsilon}{\ln d + \ln \ln \frac{1}{\varepsilon} + \ln \frac{1}{\delta}}\right)$. Then with probability at least $1 - \delta$:*

1. *The output halfspace $\boldsymbol{v}$ outputs a prediction different from $\boldsymbol{u}$ with probability at most $\varepsilon$.*
2. *The number of label queries is $O\left(d \ln \frac{1}{\varepsilon} \cdot \left(\ln d + \ln \frac{1}{\delta} + \ln \ln \frac{1}{\varepsilon}\right)\right)$.*
3. *The number of unlabeled examples drawn is $O\left(d \cdot \left(\ln d + \ln \frac{1}{\delta} + \ln \ln \frac{1}{\varepsilon}\right)^2 \cdot \frac{1}{\varepsilon} \ln \frac{1}{\varepsilon}\right)$.*
4. *The additional number of mistakes that the algorithm makes compared to $\boldsymbol{u}$ is $O\left(d \cdot \ln \frac{1}{\varepsilon} \cdot \left(\ln d + \ln \frac{1}{\delta} + \ln \ln \frac{1}{\varepsilon}\right)^2\right)$.*
5. *The algorithm runs in time $O\left(d^2 \cdot \left(\ln d + \ln \frac{1}{\delta} + \ln \ln \frac{1}{\varepsilon}\right)^2 \cdot \frac{1}{\varepsilon} \ln \frac{1}{\varepsilon}\right)$.*

The skeleton of our algorithm is adapted from that of Yan and Zhang [2017], with several key modifications to accommodate strategic behavior and a more general instance space—specifically, one where true attribute vectors are drawn uniformly from within the unit ball rather than restricted to its surface; that is, $\|\boldsymbol{z}\| \leq 1$ instead of $\|\boldsymbol{z}\| = 1$.

## 3.1 The Non-Strategic Active Perceptron of Yan and Zhang [2017]

We begin by outlining the core ideas behind the algorithm of Yan and Zhang [2017] before describing our adaptations. The algorithm has an outer and inner layer and proceeds in epochs. The outer layer is nearly identical to ours, as shown in Algorithm 1, except that it does not incorporate the manipulation cost parameter $c$. The outer layer initializes with a hypothesis vector $\boldsymbol{v}_0$ and invokes the inner layer in successive epochs, each with updated parameters such as target error, confidence level, and active learning bandwidth. The outcome of each epoch is an updated hypothesis $\boldsymbol{v}_i$, which serves as the starting point for the next. The total number of epochs is logarithmic in $1/\varepsilon$, where $\varepsilon$ is the final target error.

The inner layer of the algorithm defines both the update rule and the label-query mechanism. In the non-strategic setting, the algorithm specifies a label query region $R_t$. In the implementation of Yan and Zhang [2017], which assumes that examples lie on the surface of the unit sphere, this region consists of examples whose dot product with the current hypothesis lies in the interval $[b/2, b]$. Compared to earlier active Perceptron algorithms such as Dasgupta et al. [2005], the use of a lower bound on the dot product helps ensure that each update makes sufficient progress, thereby accelerating convergence. As for the update rule, when the algorithm makes a mistake on a queried example, it updates the current hypothesis using $\boldsymbol{w}_{t+1} = \boldsymbol{w}_t \pm 2(\boldsymbol{w}_t \cdot \boldsymbol{x}_t)\boldsymbol{x}_t$. This update rule, first proposed by Dasgupta et al. [2005], guarantees that the angle between the current hypothesis and the optimal

separator $\boldsymbol{u}$ monotonically decreases. Furthermore, it preserves the unit norm of the hypothesis vector as long as $|\boldsymbol{x}_t| = 1$.

The proof builds on the fact that, with high probability, both the angle between the current hypothesis and $\boldsymbol{u}$ and the width of the label query region shrink by a constant factor after each epoch.

## 3.2 Our Strategic Variant of the Active Perceptron

We begin by explaining the new prediction rule, which follows that of Ahmadi et al. [2021] and adjusts the classification threshold to account for manipulation costs. We then show that, under our strategic utility model, any example that is predicted negative has not been manipulated.

**Prediction Rule.** Rather than using the standard threshold $\boldsymbol{v}_t \cdot \boldsymbol{x}_t \geq 0$, our prediction rule raises the threshold to $\boldsymbol{v}_t \cdot \boldsymbol{x}_t \geq \frac{1}{c}$, where $c$ is the cost per unit of manipulation. This adjustment, following Ahmadi et al. [2021], accounts for agents' strategic behavior and ensures that, upon convergence to the optimal classifier, (the majority of the) truly positive points either lie on the positive side or can manipulate to reach it. Meanwhile, truly negative points remain on the negative side and would incur negative utility if they attempted to manipulate and be classified as positive.

To analyze agent behavior in the strategic setting, we begin by characterizing their actions under the given utility structure and prediction rule. The following result formalizes the conditions under which agents choose to manipulate their features and the resulting outcomes. In particular, it shows that examples classified as negative are guaranteed to be unmanipulated, a property that is essential for ensuring the correctness of our update rule.

**Lemma 3** (Strategic Action)**.** *Consider the following utility structure for agents, where $\|\boldsymbol{v}_t\| = 1$. Each agent receives a value of 1 if classified as positive and $0$ otherwise, and pays a cost of c per unit of movement (manipulation). The agent's utility is defined as the value received minus the cost incurred. Under the prediction rule defined above:*

*1. If $\boldsymbol{z}_t \cdot \boldsymbol{v}_t < 0$, the agent does not move and is classified negative.*
*2. If $0 \leq \boldsymbol{z}_t \cdot \boldsymbol{v}_t < 1/c$, the agent moves in the direction of $\boldsymbol{v}_t$ to a point where $\boldsymbol{x}_t \cdot \boldsymbol{v}_t = 1/c$, and is classified positive.*
*3. If $1/c \leq \boldsymbol{z}_t \cdot \boldsymbol{v}_t$, the agent does not move and is classified positive.*

**Label Query Region.** In the modified version of the algorithm, we query labels (and perform updates) only for examples that are classified as negative and lie within a specific range. We define this label-requesting region as $R_t = \left\{ \boldsymbol{x} \,\middle|\, -b \leq \boldsymbol{w}_t \cdot \hat{\boldsymbol{x}} \leq \frac{-b}{2} \right\}$. This design is essential for both addressing the strategic behavior of agents and accommodating instance vectors that are not restricted to the surface of the unit sphere. It marks a key point of departure from both classical active Perceptron algorithms and previous work on strategic Perceptron. (1) Since we focus on negatively classified examples, Lemma 3 guarantees that these examples are unmanipulated; that is, the observed vector $\boldsymbol{x}$ coincides with the true vector $\boldsymbol{z}$. This property does not hold for positively classified examples, and thus plays no role in non-strategic active learning. (2) Unlike prior work that restricts attention to examples on the surface of the unit sphere, our algorithm also queries examples from the interior. For such queries to be representative under a uniform distribution over the unit ball, it is critical that the observed (i.e., unmanipulated) examples remain uniformly distributed—something that does not hold if examples are manipulated. (3) As we show in Lemma 3, querying within $R_t$ yields uniformly distributed samples (after normalization) conditioned on being in that region and classified negative. This ensures the correctness of the updates and maintains the convergence behavior of the algorithm.

**Update Rule.** The update rule requires only minimal modifications to account for strategic behavior and a more general instance space. Since examples may lie anywhere within the unit ball, we normalize each queried example to the unit sphere by setting $\hat{\boldsymbol{x}}_t = \boldsymbol{x}_t / \|\boldsymbol{x}_t\|$. This normalization ensures compatibility with the geometric assumptions underlying the analysis and avoids distortions due to varying magnitudes. Updates are performed only on examples that are truly positive but misclassified as negative. As established in Lemma 3, such examples are guaranteed to be unmanipulated and therefore reflect the true feature vectors of the agents. The update step itself takes the form $\boldsymbol{v}_{t+1} = \boldsymbol{v}_t + 2\left(\boldsymbol{v}_t \cdot \hat{\boldsymbol{x}}_t\right)\hat{\boldsymbol{x}}_t$, which mirrors the standard active Perceptron update, except for the added normalization. This scaling step is essential for maintaining the unit norm of the hypothesis vector, which in turn ensures the correct convergence behavior of the algorithm.

---

**Algorithm 1:** Active-Strategic-Perceptron Algorithm

---

**Input:** Labeling oracle $\mathcal{O}$, initial halfspace $\boldsymbol{v}_0$, target error $\varepsilon$, confidence $\delta$, sample schedule $\{m_k\}$, band width $\{b_k\}$, manipulation cost $c$.

**Output:** Learned halfspace $\boldsymbol{v}$.

Let $k_0 = \lceil \log_2(1/\varepsilon) \rceil$.

**for** $k = 1, 2, \ldots, k_0$ **do**

   $\boldsymbol{v}_k \leftarrow$ Modified-Strategic-Perceptron$(O, \boldsymbol{v}_{k-1}, \frac{\pi}{2^k}, \frac{\delta}{k(k+1)}, m_k, b_k, c)$.

**return** $\boldsymbol{v}_{k_0}$.

---

---

**Algorithm 2:** Modified-Strategic-Perceptron Algorithm

---

**Input:** Labeling oracle $\mathcal{O}$, initial halfspace $\boldsymbol{w}_0$, angle upper bound $\theta$, confidence $\delta$, number of iterations $m$, band width $b$, manipulation cost $c$.

**Output:** Improved halfspace $\boldsymbol{w}_m$.

**for** $t = 0, 1, 2, \ldots, m-1$ **do**

   Define region $R_t = \{\boldsymbol{x} \mid -b \leq \boldsymbol{w}_t \cdot \hat{\boldsymbol{x}} \leq \frac{-b}{2}\}$.

   Observe $\boldsymbol{x}$, where $\boldsymbol{z}$ is a fresh draw from $D$.

   **while** $\hat{\boldsymbol{x}} \notin R_t$ **do**

      Predict positive if $\boldsymbol{v}_t \cdot \boldsymbol{x} \geq \frac{1}{c}$ and negative otherwise.

      Observe $\boldsymbol{x}$, where $\boldsymbol{z}$ is a fresh draw from $D$.

   $\boldsymbol{x}_t \leftarrow \boldsymbol{x}$.

   Predict positive if $\boldsymbol{v}_t \cdot \boldsymbol{x} \geq \frac{1}{c}$ and negative otherwise.

   Observe label $y_t$ of $\boldsymbol{x}_t$ by querying oracle $\mathcal{O}$.

   **if** $y_t = +1$ **then**

      Update $\boldsymbol{w}_{t+1} \leftarrow \boldsymbol{w}_t + 2(\boldsymbol{w}_t \cdot \hat{\boldsymbol{x}}_t)\hat{\boldsymbol{x}}_t$.

   **return** $\boldsymbol{w}_m$.

---

## 4 Discussion

**Inside vs. On-the-Surface Geometric Assumptions.** Much of the prior literature on active classification assumed, for simplicity and cleaner mathematical formulations, that all examples lie on the surface of a unit ball. While this assumption was inconsequential to the goals of previous research, it becomes crucial in strategic scenarios, where the relationship between observed and true features is strongly influenced by the geometry of the space. In the strategic setting, this assumption provides a straightforward case for analysis, as it allows the original unmanipulated positions of examples to be completely recovered under mild conditions. Specifically, given a linear classifier and observing an example at position $\boldsymbol{x}$, the true position $\boldsymbol{z}$ of the example on the surface of the unit ball can be recovered through an orthogonal projection, leveraging properties of the utility function.

To illustrate the robustness of our techniques, we relax this assumption—an aspect that was less relevant in prior work due to their different focus. While the direction of manipulation can still be computed based on the linear classifier in action (as the direction is always perpendicular to the classifier), the true unmanipulated position $\boldsymbol{z}_t$ is not recoverable from the observed $\boldsymbol{x}_t$ because the magnitude of the manipulation is unknown.

**Limitations and Future Work.** Our current discussion does not capture more complex forms of strategic behavior, such as agents with non-linear cost models, collusion, or asymmetric incentives. While we conjecture that some of our techniques generalize to broader settings—for example, to other smooth or approximately uniform distributions—formal extensions and analysis are left for future work. Another natural direction is to explore the robustness of active learning under different utility structures or distributional shifts, as well as the integration of fairness or strategic auditing mechanisms.

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
