# OpenReview forum: "The Active and Noise-Tolerant Strategic Perceptron"
_NeurIPS.cc/2025/Conference — Submitted to NeurIPS 2025_

### Official Review · Reviewer_ULb1 · 2025-06-23

**Clarity:** 3
**Significance:** 3
**Originality:** 3
**Rating:** 5
**Confidence:** 4

**Summary:**

This paper considers a strategic classification variant for active learning: at each time $t$, the learner picks one example $x_t$ to classify and ask for its label. The example $x_t$ observed by the learner is actually manipulated by an adversary from  $z_t$ where the goal of the adversary is to make the learner output a positive label while keeping the cost of manipulating low. Specifically, it assumes the adversary wants to maximize a utility function of $\text{PredictedLabel}(x_t) - cost(z_t, x_t)$. On the other hand, the label returned by the oracle is based on the original example $z_t$. The goal of the learner is to learn a classifier good for the original distribution of $z_t$ while making as few mistakes during the process as possible in the presence of the adversary.

In this paper, it considers a specific setting: it considers learning a homogeneous linear classifier, and assumes that that the data distribution for $z_t$ is uniform over the unit ball, the oracle follows a bounded noise model. For the adversary, it assumes $cost(z_t, x_t) = c || x_t - z_t||$.

Its algorithm is based on a perceptron-based algorithm by [Yan and Zhang, 2017], where the key insight is that due to the specific utility function of the adversary, negative examples are not manipulated, and because the distribution is assumed to be uniform, we can learn a good classifier with just these "half" examples. It proves that the label complexity of the proposed algorithm is almost the same as the classical one without strategic classification.

**Questions:**

See "Strengths And Weaknesses".

**Ethical Concerns:**

["NO or VERY MINOR ethics concerns only"]

**Final Justification:**

The author rebuttal addresses my concerns. Overall, I think it's a nice thoery paper. The downside is that its setting is a bit niche, and its assumptions are a bit strong, however, as the author explained that these assumptions are common among the research in this topic and sometimes necessary to obtain interesting results, I would not count it too heavily.

**Limitations:**

yes

**Quality:**

3

**Strengths And Weaknesses:**

Quality: This paper is technically sound. I did not check proofs in the Appendix, but the authors provided a clear explanation of their main technique, which looks good to me and should be reasonable to achieve the claimed theoretical results.

Clarity: This paper is mostly clear and easy to follow. One minor suggestion is that the setting would be clearer if the authors could clearly (re)state in each iteration, which example the learner selects, what the learner observes, what the example is actually is in the hindsight, what the adversary knows and does, and according what the oracle returns a label.

Significance: This paper considers an interesting question and provides a solid algorithm with theoretical guarantees for it. One weakness is that it looks like the result is very specific to the assumption of the utility function and data distribution, and it would be helpful if the authors could explain if these assumptions are necessary or common in the literature, and if the results can be generalized to other utility functions or data distributions.

Originality: The main algorithm is based on an earlier work of [Yan and Zhang, 2017], but it looks nontrivial to adapt and apply it to solve the problem studied in this paper.

---

> ### Author Rebuttal · Authors · 2025-07-31
>
> Thank you for your thoughtful review. We address your suggestions and questions below.
>
> 1. Clarifying the iterative setting.
>
> Thank you for this suggestion. We will add a short summary in Section 3.2 explaining, for each iteration, which example the learner selects, what is observed, what the true example is, what the agent does, and how the oracle returns the label. This will make the process easier to follow.
>
> 2. Assumptions and generalization.
>
> You are correct that our guarantees depend on the assumptions of the utility function and data distribution. These assumptions (linear cost utility and uniform/smooth data distributions) are standard in the active learning and strategic classification literature (e.g., Hardt et al. 2016; Yan & Zhang 2017; Ahmadi et al. 2021) and allow us to obtain tight, exponential label savings. We will add text clarifying why these assumptions are necessary for our analysis and how they can be generalized:
>
> * The uniform distribution assumption can be relaxed to smooth generalizations where density ratios along rays are bounded (as outlined in Section 1.3 and Appendix B).
>
> * Our approach can also handle unknown costs. We will add this to the revised version.
>
> 3. Novelty relative to Yan & Zhang (2017).
>
> While our algorithm is inspired by Yan & Zhang (2017), adapting it to the strategic setting is non-trivial. We had to address manipulation in feature space, identify unmanipulated points for reliable updates, and extend the analysis from the unit sphere to the unit ball using new coupling arguments. Furthermore, we do not view the similarity of our algorithm to Yan & Zhang (2017) as a downside; on the contrary, it is both surprising and appealing that nonsignificant changes to the algorithm can yield strong guarantees in the strategic setting. We will make these distinctions clearer in the introduction and Section 3.

---

> > ### Comment · Reviewer_ULb1 · 2025-08-07
> >
> > Thanks for the clarification. Can you elaborate a bit more regarding why "it is both surprising and appealing that nonsignificant changes to the algorithm can yield strong guarantees in the strategic setting"?

---

> > > ### Author Response · Authors · 2025-08-08
> > >
> > > Thank you for the follow-up question.
> > >
> > > Strategic behavior often breaks key assumptions that traditional algorithms rely on, such as observing i.i.d. samples or truthful inputs, so even small changes to the model or objective can significantly complicate learning. What makes our result surprising is that, despite this, simple and clean modifications recover the core guarantees of active learning, even when agents adapt their features in response to the classifier.
> > >
> > > It is appealing because the final algorithm remains transparent and efficient, and yet is robust to strategic behavior without relying on heavy machinery or retraining with counterfactual simulations. We find it surprising that such a minimal and interpretable adjustment suffices to recover exponential improvements in label complexity, which are often lost in more complex or adversarial settings.

---

### Official Review · Reviewer_gTUt · 2025-06-30

**Clarity:** 1
**Significance:** 2
**Originality:** 1
**Rating:** 4
**Confidence:** 3

**Summary:**

This paper studies a strategic classification variant (meaning data points, aka users, can manipulate their features to get a positive label) of the active perceptron. The authors study the realizable and nonrealizable case (with bounded noise) under strong assumptions (uniform distribution over the sphere, noise is not too severe). They essentially recover the query complexity of the active perceptron of Yan and Zhan [2017] while dealing with the more involved strategic classification setting [Ahmadi et al. 2021]. The latter being non-active requires significantly more labels.

**Questions:**

You briefly mention margin-based active learning but the relationship is not fully clear. Can the standard margin-based analysis of the perceptron or the corresponding active variants be used here as well? In some sense is the region, where the users will flip from negative to positive, strongly corresponding to the margin (perhaps a one-sided version of it) in standard learning of halfspaces.

The assumption on $\eta$ in the theorem (while being the same as by Yan and Zhang) is not nice at all with $eta$ depending on all relevant parameters in the paper (including $\delta$ and $d$). Can you rephrase the theorem in a way that this condition only relates $\eta$ and $\epsilon$? This would be much more natural. Either a specific $\epsilon$ is achievable or the noise is simply too high (see e.g., Hanneke and Yang 2015, Theorem 8).

It is claimed that the uniform on the surface case is easy for the strategic case. Please add a more detailed discussion/proof.

**Ethical Concerns:**

["NO or VERY MINOR ethics concerns only"]

**Final Justification:**

The authors addressed various concerns. While I still belief that the paper is not in the best shape and would benefit from a thorough rewriting, I raised my score to a borderline accept.

**Limitations:**

The authors do not really discuss the limitations of say the strong assumptions of the uniform distribution on the sphere. This is however not too uncommon in active learning. They do discuss the limitation of assuming a good initial guessed halfspace in Appendix C with a simple solution. For some reason this is absolutely not clear from the main text and not referenced there.

**Paper Formatting Concerns:**

.

**Quality:**

2

**Strengths And Weaknesses:**

This paper addresses a timely and important combination of strategic classification and active learning. As far as I am aware this is the first paper tackling both problems simultaneously.

There are multiple bigger weaknesses:

1. This submissions seems to be a rather small step on top of Yan and Zhang (2017). In particular, inspecting the appendix we see that the most lemmas leading to the main result are either small adaptations from either Yan and Zhang or Ahmadi et al (2021). The main result itself is proven by reducing the strategic classification case to the standard non-strategic case (which is not really clear from the main paper). The main ingredient to handle the strategic case seems to be very close to the Ahmadi et al (2021) strategy. So it is rather unclear what the main contribution of this paper here is.

2. The writing and structure is not great. The main theorem is presented at page 6, the main algorithm only at page 8. The whole paper mixes the authors' contributions and discussion of previous work, making it hard to judge its contribution. Moreover, the main paper even makes it seem that the contributions are more limited than they are. In the main paper it seems like they require an initial halfspace which is already quite close to the best halfspace. However in the not referenced Appendix C they discuss a simple strategy by Yan and Zhang (2017) to alleviate this issue. The authors have a full more page left in the main paper (they only have 8 pages), where they could have added such details.

3. The assumptions are really strong: uniform distirbution on the sphere, an assumed noise level which depends on all relevant parameters, and the initial good halfspace (which however is resolved in the Appendix C as mentioned above). While all these assumptions come from Yan and Zhang (2017), one could argue (or might hope/expect) that during these 8 years more sophisticated strategies/analysis have been developed for broader families of distributions. I am not sure it is sufficient to just repeat the limitations from previous papers.

Smaller comments:
* the cost$(\cdot,\cdot)$ and the whole utility value$(x)-$cost make this paper seem more general then it is. Consider putting the actually used weighted Euclidean distance and the three cases of the utily (basically like in Lemma 3/its proof).

---

> ### Author Rebuttal · Authors · 2025-07-31
>
> Thank you for your constructive review. We address your points below.
>
> **Perceived small step over Yan & Zhang (2017).**
>
> Our contributions are orthogonal and non-trivial:
>
> 1. We adapt active learning guarantees to a setting with strategic manipulation, a key departure from non-strategic analysis.
>
>
> 2. We identify a structural property unique to strategic classification: only negative-classified points are guaranteed unmanipulated and thus safe for updates.
>
>
> 3. We extend the analysis from the unit-sphere to the unit-ball distribution.
>
>
> 4. We provide noise-tolerant guarantees in the strategic setting, addressing an open problem from Ahmadi et al. (2021).
>
> We would like to emphasize that our contribution is primarily conceptual rather than introducing a highly complex technique. We do not view it as a drawback that our results are obtained through relatively simple modifications of existing algorithms and hence following a similar proof structure; on the contrary, it is both surprising and appealing that small, clean changes can transform strong guarantees to the strategic setting.
>
> **Writing and structure.**
>
> We will: (i) present Algorithms 1 and 2 earlier, before Theorem 2; (ii) include proof sketches in the main text; (iii) bring the Appendix C discussion on relaxing the initial-halfspace assumption into the main text; and (iv) better separate prior work from our contributions.
>
> **Assumptions.**
>
> We acknowledge that the uniform distribution, bounded noise, and initial halfspace assumptions are strong. These are standard in active learning theory. We now clarify that:
>
> * our arguments extend to smooth non-isotropic distributions (Section 1.3, Appendix B);
>
>
> * the initial halfspace assumption can be removed using a bootstrapping strategy (Appendix C, to be referenced in the main text);
>
>
> * the noise tolerance we achieve is essentially optimal for this class of algorithms.
>
>
> **Margin-based connection.**
>
> We believe our techniques work for margin-based active variants of the perceptron algorithm. In fact, we first verified this for the variant from Dasgupta et al. (2005), but then we realized that if we use Yan and Zhang (2017)’s algorithm, our final algorithm will be cleaner and the proof will be much simpler. We will explicitly highlight how the margin-based analysis is adapted to the strategic setting.
>
> **Noise parameter condition.**
>
> Thank you for the suggestion. We will rewrite the noise condition more intuitively: if $\nu < \nu^\star(\epsilon, d)$ the algorithm succeeds; if $\nu > \nu^\star$, no active learner can guarantee error $\epsilon$ (Hanneke & Yang 2015).
>
> **Uniform on the surface.**
>
> In the case where points are uniformly distributed on the surface of the unit sphere, we can show that there is a one-to-one mapping between each observed point and its original point. To illustrate, consider the two-dimensional case where points lie on the circumference of a circle and the linear classifier is positioned at a distance $1/c$ from the origin. Half of the points (those on one semicircle) are too far to manipulate and will be classified as negative. A portion of the remaining points (on the opposite arc) already lie on the positive side of the classifier and are classified as positive without manipulation. The rest will manipulate to exactly reach the classification boundary; these points form two symmetric line segments, and their observed positions on the boundary correspond directly to their projections from the original points. This projection establishes a one-to-one mapping, allowing the original points to be recovered exactly.
>
> In contrast, when points are uniformly distributed inside the unit ball, this projection is no longer one-to-one but instead many-to-one. We will include additional details and formal explanations of this distinction in the revised version.

---

### Official Review · Reviewer_SSTP · 2025-06-30

**Clarity:** 2
**Significance:** 1
**Originality:** 2
**Rating:** 3
**Confidence:** 3

**Summary:**

In the present work, the authors study active learning in strategic-classification settings, where agents can manipulate their features (inputs at inference) to be classified positively. Building on Yan & Zhang (2017) and Ahmadi et al. (2021), the authors adapt the Active Perceptron algorithm:

1- to raise the decision threshold from $0$ to $1/c$ (where $c$ is the per-unit manipulation cost);

2- query labels only for points currently classified as negative and lying in a narrow “band” around the hyperplane;

3- normalise queried points and update with a Perceptron-style rule.


For data drawn uniformly from the $d$-dimensional unit ball (not just the sphere) they assert (without proof) that, under a bounded-noise condition, the modified algorithm achieves excess error $\epsilon$ with $\mathcal{O}(d\log \frac{1}{\epsilon})$ queries, incourring in the same amount of mistakes.

**Questions:**

1- Practical motivation: Can you name a concrete application where the hypotheses required for this study are close enough to realistic assumptions? For example, settings where the $1/c$ threshold employed by the agents is known by the learner, or where the isotropic feature assumption is close to true?

2- Noise condition: How does the bound on the admissible noise rate compare numerically with the passive-learning lower bound of Ahmadi et al.?

3- What does the notation $\Theta$ indicatem in Theorem 2?

4- Role of $c$: Could the algorithm be run with a mismatched $c$?

5- Uniform-ball assumption: Given the presence of many other simplifying assumptions, why is relaxing the sphere normalization important?

6- Can the authors provide any empirical validation of the effectiveness of their approach, especially when some of their hypotheses might be violated?

**Ethical Concerns:**

["NO or VERY MINOR ethics concerns only"]

**Final Justification:**

I maintain my skepticism about the broadness of the impact of this work. Taken as a merely theoretical work, it seems that the novelty is limited when compared to the closely related cited work. Taken as a work introducing a novel algorithm, the unclear applicability of the setting in real world problems, the reliance on unrealistic hypotheses, and most importantly the lack of empirical validation on real data hinders the solidity of the results. If this type of approach is typical of this sub-field, maybe this work is bettered positioned to appear in a more specialized conference.

**Limitations:**

No empirical evidence: even a small-scale simulation could demonstrate behaviour when assumptions are violated (e.g., non-isotropic data, heterogeneous costs).

Novelty gap: The algorithm is closely related to two prior works, and the technical details are mostly inherited straightforwardly.

**Paper Formatting Concerns:**

The maximum length was not exploited, although the paper is missing important empirical validation.

**Quality:**

2

**Strengths And Weaknesses:**

**Strengths**
- Clear high-level motivation for reducing label complexity in costly domains.
- Opens an under-explored combination of active learning and strategic manipulation, claiming exponential label savings in strategic classification.

**Weaknesses**
- No proof sketch of the main theorem is supplied in the main; in the appendix, the proof is shown to be derivative of Yan and Zhang (2017).
- Not clear why Lemma 3 is presented as a Lemma, since it is a restatement of the prediction rule and the action of the strategic agents.
- Theoretical setup assumes agents know and use the same $1/c$ threshold that the learner uses; it is not fully clear why these thresholds would coincide and why the learner would know this threshold.
- Uniform-ball vs uniform-sphere generalisation is presented as a strong point of this work, but its impact is not discussed convincingly. In high dimensions, for example, the difference is negligible.
- The length constraint was not an issue in this work, since page 9 was not utilized (and Section 2 is often redundant). However, no empirical sanity checks are offered, even though the algorithm’s robustness to violated assumptions is unknown.
- Little evidence is presented on the fact that the insights provided by this work transfer to realistic pipelines.
- Heavy reliance on Yan and Zhang (2017) for analysis structure and on Ahmadi et al. (2021) for the strategic threshold; the degree of novelty of this work seems limited.

---

> ### Author Rebuttal · Authors · 2025-07-31
>
> Thank you for your review. Below we respond to your questions and comments.
>
> **Q1. Practical motivation and realism of assumptions.**
>
> Our framework applies to domains where strategic manipulation and costly labeling coexist, such as hiring and admissions (applicants modify resumes, labels require expert review), financial systems (credit scores can be manipulated, labels such as default events are expensive), and platform moderation (users adapt behavior, human labeling is costly). The cost parameter $c$ models known manipulation penalties (e.g., exam fees, regulation costs) and is standard in prior work (Hardt et al. 2016; Ahmadi et al. 2021). Learners can estimate $c$ from observed shifts or domain knowledge. While we assume isotropy for theoretical guarantees, our arguments extend to smooth non-isotropic distributions where density ratios along rays are bounded. We will clarify this generalization. Also, please see our response to Q4 that clarifies the algorithm can be made robust to unknown $c$ with a simple modification.
>
> **Q2. Noise condition compared to Ahmadi et al. (2021).**
>
> Our admissible noise level is
>
> $$
> \nu = O\left( \frac{\epsilon}{\ln d + \ln \ln (1/\epsilon) + \ln (1/\delta)} \right),
> $$
> matching the best-known active Perceptron bounds (Yan & Zhang 2017). This is strictly stronger than Ahmadi et al. (2021), which fails for even a single noisy data-point in the data stream. Our algorithm thus achieves exponential label savings with noise tolerance, closing an open gap.
>
> **Q3. Notation in Theorem 2.**
>
> The $\Theta$ bounds merely drop the constant multipliers. We will clarify this explicitly.
>
> **Q4. Robustness to mismatched $c$.**
>
> Our algorithm uses $c$ as an input; however, it is relatively straightforward to determine whether $c$ is being overestimated or underestimated, and by how much, by examining unlabeled samples. In the case of a uniform distribution (or smoothed generalizations), if we are overestimating $c$, we will observe data points in a margin region around the origin on the positive side. This occurs because the classification threshold is set too conservatively, preventing many points that could otherwise move across the boundary from doing so. Conversely, if we are underestimating $c$, the margin region around the origin on the negative side will be essentially empty, as manipulated points can more easily cross the boundary. By conditioning on the observation of points in these margin regions, one can infer whether $c$ is being over- or underestimated and adjust it accordingly.
>
> **Q5. Uniform-ball assumption.**
>
> The uniform-ball setting is strictly harder than the unit-sphere case because true features have varying norms, and we cannot exactly recover unmanipulated points. If we assume points are on the surface, in our setting we are able to fully recover the original point. We will clarify why this extension matters.
>
> **Q6. Empirical validation.**
>
> We appreciate the suggestion. Our focus is a theoretical contribution, but empirical validation is a valuable future direction.
>
> **Novelty.**
>
> Although we build on Yan & Zhang (2017) and Ahmadi et al. (2021), our conceptual contributions are substantial:
>
> 1. first to bring active learning to strategic classification,
>
>
> 2. identifying how to learn solely from unmanipulated negative-classified points,
>
>
> 3. extending the analysis from the sphere to the ball,
>
>
> 4. closing the noise-tolerance gap for the strategic setting.
>
>
>  We will make these points clearer in the revised version.
>
> We would like to emphasize that our contribution is primarily conceptual rather than based on introducing a highly complex technique. We do not view it as a limitation that our results are obtained through relatively simple modifications of existing algorithms; on the contrary, it is both surprising and appealing that small, clean adaptations can yield strong guarantees in the strategic setting. The simplicity of our algorithm is a natural consequence of selecting the most compatible active learning method from several possible alternatives in the literature. Our techniques can also be applied to other active variants of the Perceptron algorithm (e.g., Dasgupta et al., 2005). We chose Yan and Zhang (2017) as our baseline to highlight how strong guarantees in the strategic case can be achieved with small but carefully designed adaptations.

---

> > ### Comment · Reviewer_SSTP · 2025-08-05
> > **Comment**
> >
> > I would like to thank the authors for their responses. I have a few follow-up questions:
> >
> > > Q1
> >
> > Can the authors provide an example of a real-data benchmark where their hypotheses apply?
> >
> > > Q4
> >
> > It seems that these considerations require the strategic agents to solve the minimization problem stated in the first equation in the paper exactly. But what if their behavior is not optimal, and therefore not as predictable? Wouldn't this be a realistic case if learners are to "estimate $c$ from observed shifts or domain knowledge"?
> >
> >
> > > Q6 + We would like to emphasize that our contribution is primarily conceptual rather than based on introducing a highly complex technique. We do not view it as a limitation that our results are obtained through relatively simple modifications of existing algorithms; on the contrary, it is both surprising and appealing that small, clean adaptations can yield strong guarantees in the strategic setting.
> >
> > I am not fully convinced that, in any realistic strategic setting, the proposed algorithm would be quite as effective as in the controlled theoretical setup proposed in this analysis. The authors are assuming that: (1) the input distribution is known, (2) the strategic behavior is informed, (3) each agent applies the same minimal/optimal strategy (e.g., everyone uses the same $c$), so that it is completely predictable what they would/would not do. I understand that, in this setting, the negative points are then guaranteed to be informative. However, I do not understand why there wouldn't be sources of obfuscation in most real settings. This is precisely why any empirical validation (which is often present also in studies of a theoretical nature) would help clarify the robustness of this simple algorithm.

---

> > > ### Author Response · Authors · 2025-08-08
> > >
> > > Thank you for the follow-up questions. We address them below.
> > >
> > > Q1.
> > >
> > > We refer the reviewer to the datasets used in “Strategic Classification Made Practical” by Levanon and Rosenfeld, and “Steering User Behavior with Badges” by Anderson, Huttenlocher, Kleinberg, and Leskovec.
> > >
> > > Q4.
> > >
> > > You are correct that our analysis assumes agents follow the minimal-cost manipulation strategy. Theoretically, rational behavior is one of the most common assumptions in strategic settings. In practice, strategic behavior may be noisy or suboptimal due to limited information or bounded rationality. In our algorithm, such suboptimality would translate into additional noise. However, the labeling and update procedures rely only on the fact that negative-labeled examples are unmanipulated—a property that holds even when behavior is not fully optimal. This makes our algorithm robust to deviations from perfect rationality, as the updates are still based solely on reliably unmanipulated inputs, with suboptimal behavior introducing only an additional source of noise.
> > >
> > > Q6.
> > >
> > > The assumptions we make are standard in many of the influential papers we cite. While we agree that empirical validation can help demonstrate the robustness of our simple algorithm, our primary focus in this work is on the theoretical analysis, in line with prior research in this area.

---

### Note · Authors · 2025-08-15

We would like to take this opportunity to address the area chair’s questions and further emphasize the novelty and significance of our work.

Thank you for the thoughtful questions. Below, we address each point in turn.

**1. On whether our results could be improved to $O(\epsilon)$ noise tolerance via mirror descent (Shen, ICML 2021)**

We believe our ideas have the potential to be extended by replacing the Perceptron with localized online mirror descent, as in Shen (2021), which may improve noise tolerance from $O(\epsilon/\log d)$ to $O(\epsilon)$. Formally establishing this remains an open question for future research, but it highlights the strength and flexibility of our approach.

Intuitively, both our algorithm and Shen’s use one-sided margin updates, and our queried points are all negative and unmanipulated, allowing Shen’s analysis to largely carry over. We would retain the same band sampling and strategic threshold, swap in OMD updates, and potentially achieve the improved bound without increasing label complexity.

More broadly, our work is the first to combine active learning with the strategic classification setting. This framework has the potential to be applied to a broad class of active learning algorithms—which we believe opens an exciting new research direction.

**2. On the noise threshold from Hanneke & Yang.**

Our wording was imprecise–we are not making any specific claims about their results.

**3. Why malicious noise results (e.g., Awasthi–Balcan–Long, JACM 2017) do not apply**

The malicious noise model assumes that only a fraction (typically $O(\epsilon)$) of examples are corrupted arbitrarily, while the rest are drawn cleanly from the target distribution. In our strategic setting, **every** example **may be** strategically manipulated; of course if the manipulation is completely arbitrary, we cannot hope to prove much. Following the line of work in strategic classification, we assume that the manipulation is *structured*—agents move their features to optimize their own utility given the classifier. This is qualitatively different from arbitrary corruption (as done in the malicious noise model): the noise is distribution-dependent and coupled with the learner’s current decision boundary.

As a result, algorithms designed for malicious noise rely heavily on the presence of a large fraction of uncorrupted examples in each round, a property our setting does not guarantee.

We will incorporate this discussion into the final version.

---

### Decision · Program_Chairs · 2025-09-17

**Decision:**

Reject

**Comment:**

The submission receives a mixed rating throughout the reviewing process. On the positive side, reviewers found that the paper is written well and the algorithm is easy to follow. On the flip side, reviewers agree that the setup (such as distributional assumptions and very specific utility and cost functions) is very limited. Some reviewers also raised the concern that the key techniques to achieve label efficiency and noise tolerance - which are claimed as main contributions - have been set out in Yan & Zhang (and have been well-known techniques in active learning). In addition, the algorithmic ideas that modify Perceptron for strategic agents have also been set out in prior works such as Ahmadi et al.

Overall, as a learning theory paper, the technical contribution does not meet the bar.